# Regional Temporal and Spatial Trends in Drought and Flood Disasters in China and Assessment of Economic Losses in Recent Years

**Jieming Chou** [1], **Tian Xian** [1,*], **Wenjie Dong** [2] and **Yuan Xu** [1]

1   State Key Laboratory of Earth Surface Processes and Resource Ecology, Faculty of Geographical Science, Beijing Normal University, Beijing 100875, China; choujm@bnu.edu.cn (J.C.); 201721480034@mail.bnu.edu.cn (Y.X.)
2   School of Atmospheric Sciences, Sun Yat-Sen University, Guangdong 510275, China; dongwj3@mail.sysu.edu.cn
*   Correspondence: xiant13@lzu.edu.cn

**Abstract:** Understanding the temporal and spatial distribution in disasters plays an important role in disaster risk management. The present study aims to explore the long-term trends in drought and floods over China and estimate the economic losses they cause. A peak-over-threshold approach is used to identify flood peaks, and the relationship between the disasters and climate indices is investigated using Poisson regression. The major results are as follows: (1) the northeastern part of China was severely affected by drought disasters (average damaged area was 6.44 million hectares); (2) the northern part of East China and Central China upstream of the Yangtze River were severely affected by flood disasters (average damaged area was 3.97 million hectares); (3) in the Yangtze River Basin, there are increasing trends in terms of drought and extreme precipitation, especially upstream of the Yangtze River, accompanied by severe disaster losses; and (4) by combining the trends in drought and extreme precipitation days with the spatial distribution of damaged areas, the study indicates that the increasing trend in droughts has shifted gradually from north to south, and the increasing trend in extreme precipitation gradually has shifted from south to north.

**Keywords:** Damaged area; direct economic loss; disaster; drought; extreme precipitation

---

## 1. Introduction

With global warming, the numbers of extreme precipitation events may increase in many areas [1–3]. Increased frequency of extreme events leads to increased frequency and intensity of floods and droughts, and extreme events have adverse effects on human society and the environment [4,5]. According to statistics of the World Meteorological Organization (WMO), there were 7870 meteorological disasters in the world between 1970 and 2009, which caused 1.86 million deaths and $195.4 billion in direct economic losses [6]. Analyzing economic losses in disasters can help us understand the relationship between the ecosystem and disasters and provide guidance for people to better cope with disasters. Therefore, studies about extreme events have attracted more attention [7–9].

The global average surface temperature has increased since the mid-20th century [10]. China has also experienced significant temperature increases concurrent with global warming; the projections for China indicate that the temperature is projected to increase by 1.78–5.78 °C, with larger warming over northern China and the Tibetan Plateau by the end of the 21st century [11]. Increased atmospheric greenhouse gases warm the surface but they also exacerbate surface evaporation and enhance the

ability of the atmosphere to retain water vapor, which means that the water content may increase in the atmosphere [12]. Droughts are more likely to occur when water vapor evaporation increases, and to balance the evaporation process, precipitation will also increase, and floods will occur. Therefore, precipitation has also changed because of changes in temperature [13].

Drought is recognized as the costliest and most pressing natural hazard in world, influencing water resource systems, agricultural production, and natural ecosystems [14]. Since the late 20th century, a warming trend has been observed for most parts of the world [15,16], and the warming will continue under different climate change scenarios [17]. Due to global warming, unprecedented intense climate events (e.g., floods and droughts) might occur [18]. Severe droughts in human-dominated environments, as experienced in recent years in California, Brazil, China, Spain, and Australia, cannot be seen as purely natural hazards [19]. Based on past drought events [20], China is prone to costly drought impacts, with an average annual drought-affected crop area of 209,000 km$^2$ and annual direct economic losses of more than 32 billion yuan (according to 2013 price levels) in the period 1949 to 2013. Drought affects almost everywhere, and the potential change is still a major problem, which has a significant effect on national ecological security [21]

Previous studies have investigated the changes in extreme precipitation events in recent years [22]. It has been shown that extreme precipitation increased significantly in the Yangtze River region, Southwest China, and South China from 1951 to 2000 [7], with significant increases in heavy rainfall at rural and urban stations in East China [23]. The physical interpretation of extreme precipitation has suggested that the western North Pacific subtropical high and mid-latitude wave systems have greatly affected extreme summer precipitation in China [24]. According to Orsolini [4], the Silk Road and polar waves play key roles in regulating extreme precipitation in the north of China and Northeast China. According to regional climate models, the increasing rainstorm days in parts of China is due to the greenhouse effect [25], and the effects of global warming (rather than aerosols) are considered to be responsible for the heavy rain in eastern China [23].

There are also several studies focused on drought and flood forecasting and estimating the likely drought conditions in the future. In an earlier study, Karl [26] used the unconditional gamma distribution to obtain the probabilities of future droughts. Since then, several other methods have been developed and tested in drought and flood forecasting such as Markov Chain model [27], stochastic renewal models [28,29], stochastic autoregressive models [30], and artificial neural networks [31]. Consequently, a reliable forecast model for a region has a significant role in the disaster risk management.

According to recent statistics [32,33], the economic losses caused by global climate change and related extreme climate events have increased by an average of 10 times in the past 40 years. Meteorological disasters have accounted for more than 70% of all natural disasters in China [34]. Climate change has a direct effect on increasing the frequency and intensity of extreme events such as droughts and floods, especially in climate-sensitive and vulnerable areas. Since the 1950s, the economic losses caused by droughts and floods have accounted for 78% of all meteorological disasters [35]. To reduce the damage, many studies have investigated drought and flood distribution and proposed solutions [36]. Most of these studies focused on single disasters [37,38], whereas relatively few considered multi-hazard superposition effects. Therefore, the present study used the latest and most unique data to study drought and floods and considered the characteristics of the East Asian Summer Monsoon Index (EASMI) and the Nino 3.4 Index to comprehensively analyze the characteristics of drought and extreme precipitation and the resulting direct economic losses. Thus, the research aims to provide better scientific evidence and suggest countermeasures for mitigating and managing the risk of disasters.

## 2. Data and Methods

### 2.1. Data

The study used total agricultural output value statistics for various regions from 2000 to 2003 compiled by the National Bureau of Statistics, as well as the drought damaged areas and total planted areas in various regions from 2000 to 2003 from Chinese Agricultural Statistics. The direct economic losses from 2004 to 2015 were derived from the China Meteorological Disaster Yearbook. Daily and monthly precipitation records from 855 stations in China during 2000–2015 were used (Figure 1). This dataset was obtained from the National Meteorological Administration. Relatively strict quality control was applied before use and stations with short or missing data sequences were deleted. To explore the mechanisms responsible for droughts and floods, the study also employed the EASMI (http://ljp.gcess.cn) and the Nino 3.4 Index specified by the National Climate Center for Research. For convenience, the study area (mainland China) is divided into Northeast China, North China, East China, Central China, South China, Northwest China, and Southwest China (Figure 2).

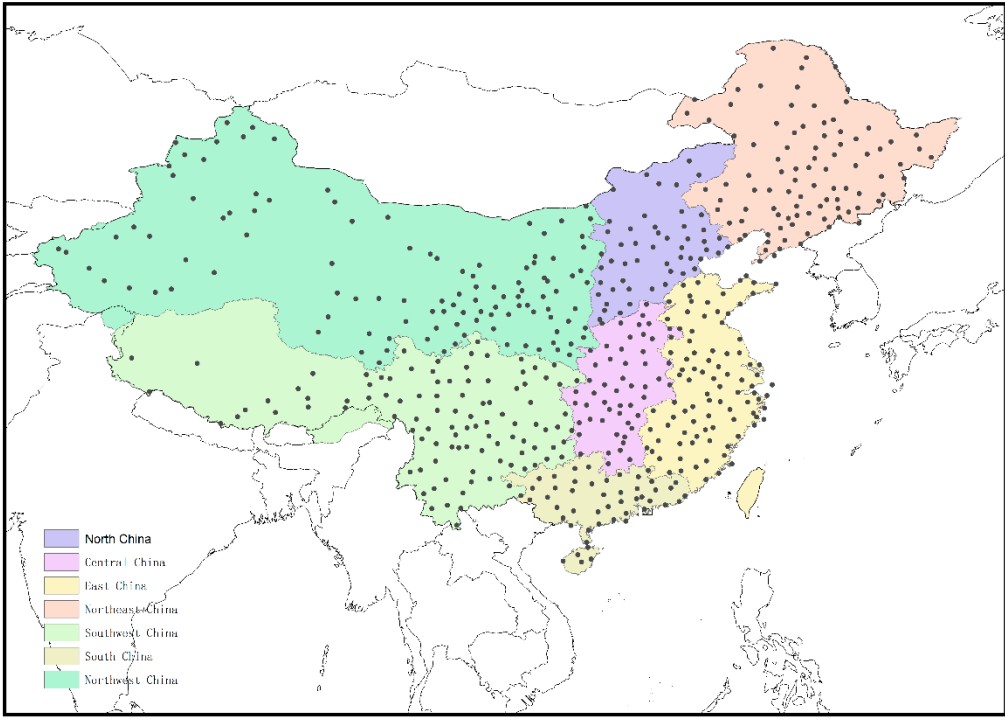

**Figure 1.** Meteorological stations in China.

### 2.2. Methods

#### 2.2.1. Drought Indicators

The study used the standardized precipitation index (SPI) as an indicator of drought [39,40]. Based on the monthly precipitation data obtained from 652 weather stations in the study area covering 16 years, the research analyzed the temporal and spatial features of meteorological drought in the whole research areas by calculating annual time scale SPI (12-month SPI). An SPI value > 1 indicates precipitation higher than the mean value in the same period, where the flooding is more severe when the value is larger. An SPI value < −1 indicates that the precipitation is lower than the average level in the same period, where the meteorological drought is more severe when the value is smaller. The representative drought indicators used throughout the world include the SPI and Z-index. Previous studies indicated that the SPI is more stable than the Z-index (unlike the SPI, the Z-index directly normalizes the probability density function) and can meet the requirements of different time scales and

analyses of the status of different water resources [41]. A 12-month SPI is a comparison of precipitation for 12 consecutive months with that recorded in the same 12 consecutive months in all previous years of available data and reflects long-term precipitation patterns [42].

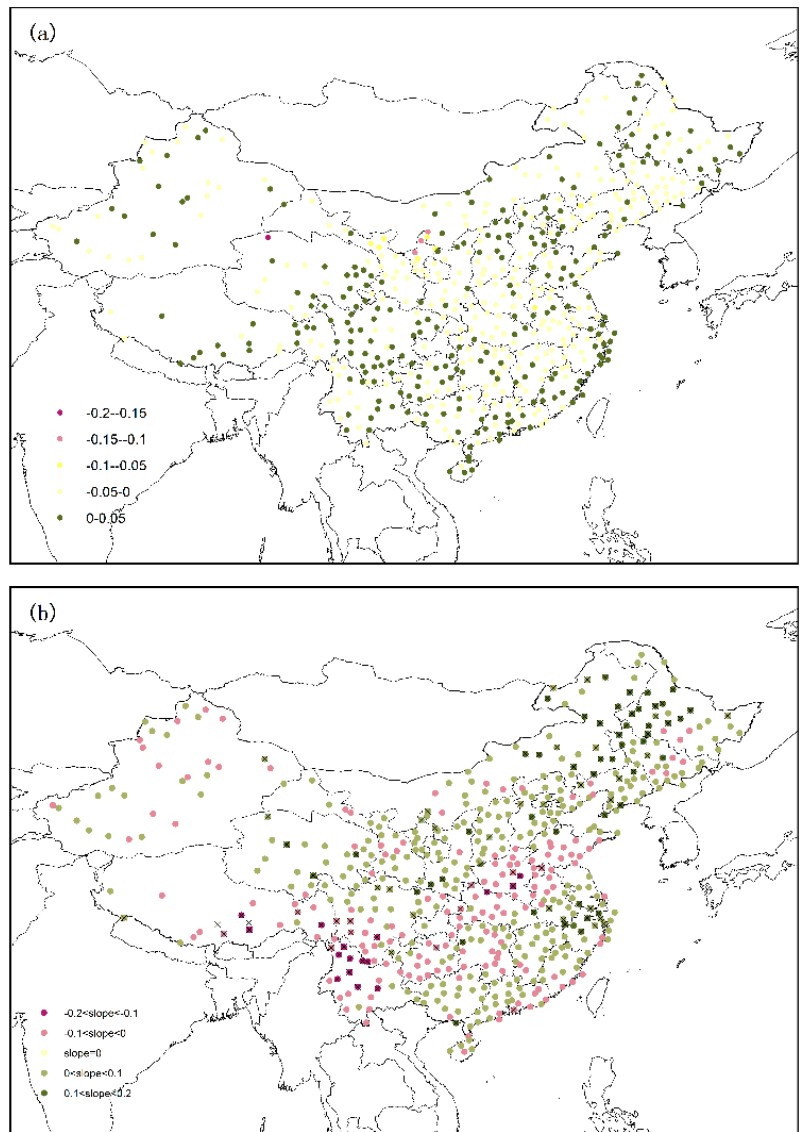

**Figure 2.** Standardized precipitation index (SPI) distribution: (**a**) annual average SPI; (**b**) SPI trend (unit: year$^{-1}$). Red dots indicate that the slope is negative, and the drought tends to increase; green dots indicate that the slope is positive, and the drought tends to decrease. The point of the cross passed the 95% significance test

### 2.2.2. Extreme Precipitation Indicator

Frich [43] compiled a new global dataset of derived indicators to clarify whether frequency and severity of climatic extremes changed during the second half of the 20th century. Among them, 5 indicators related to extreme precipitation events were selected: R10(number of days with precipitation $\geq$10 mm d$^{-1}$), CDD (maximum number of consecutive dry days ($R_{day} < 1$ mm)), $R_{5d}$ (maximum 5 d precipitation total), simple daily intensity index (SDII: annual total/number of $R_{day} \geq 1$ mm d$^{-1}$), and R95T (fraction of annual total precipitation due to events exceeding the research period 95th percentile).

In fact, the definitions of extreme precipitation events vary among places. For example, heavy rainfall and heavy rain have never occurred in the history of some stations in Northwest China, but moderate rain often causes landslides and other hazards in this area, and thus defining thresholds according to the criteria for heavy rainfall to study extreme precipitation events has no practical significance. For this reason, a percentile-based extreme precipitation index is defined [44]. For a given station, an extreme precipitation event is defined as daily precipitation beyond the 95th percentile threshold of all rainy records for all 16 years from 2000–2015.

*2.3. Estimation Method*

Due to the lack of data for direct economic losses from drought in 2000–2003, the reducing production coefficient method [45–47] was employed to estimate the losses in these years. This method measures the economic benefits or losses due to changes in environmental quality by considering changes in the output value or profit in an area attributable to changes in environmental quality. This method has often been used to calculate direct crop production losses, economic forest output value losses, timber forest losses, economic losses of grassland resources, and economic losses of fisheries [48]. When the method is used to calculate agricultural losses, it is sometimes called the reducing production coefficient method. This method employs the areas with disasters over the years to estimate agricultural losses and converts this into the grain output value. Thus, the study defined the direct loss due to drought as:

$$Q = \alpha \cdot A \cdot P_q \tag{1}$$

where $\alpha$ is the agricultural production yield reduction rate, $P_q$ is the total output value/planted area with crops in each area, and A is the area affected by the disaster.

*2.4. Analysis Methods*

2.4.1. Empirical Orthogonal Function (EOF)

The EOF was used to analyze the direct economic losses due to droughts and floods in China, and the temporal and spatial characteristics of the damaged areas. The EOF is a common method for analyzing field sequences [49]. It decomposes a certain element field sequence ($F_{ij}$) into an orthogonal time function ($T_{ih}$) and an orthogonal spatial function ($X_{hj}$):

$$F_{ij} = \Sigma \left( T_{ih} \times X_{hj} \right), \tag{2}$$

where $F_{ij}$ is the factor field; $i = 1, 2, \ldots, m$ is the time sequence number; $j = 1, 2, \ldots, n$ is the station number; $X_{hj}$ is the spatial function; $h = 1, 2, \ldots, H$ is the decomposed number of fields; and $T_{ih}$ is a function of time. The spatial function $X_{hj}$ is usually regarded as a typical field, changing completely with space. However, the time function $T_{ih}$, which only changes with time, is considered to be the weight coefficient for a typical field. The EOF generally converges quickly, where the sum of the first few typical fields can represent the actual field, and changes in the time coefficients reflect the importance of each typical field at different times. The data comprising direct economic losses and damaged areas due to drought and floods met the requirements for EOF analysis. EOF analysis can determine the basic form of the spatial distribution and the time-varying characteristics of drought and flood disasters, thereby helping to understand drought and flood disasters in China.

2.4.2. Mann–Kendall Test

The nonparametric Mann–Kendall (MK) test was used to characterize the trends of SPI and extreme precipitation days in this study. The MK testis widely used to detect trends in meteorological data [50]. The study estimated temporal changes of the SPI and time series of extreme precipitation days by calculating MK's slope estimator, and estimated trend significance using the MK test.

*2.5. Correlation Analysis*

Two climate indices (EASMI and Nino 3.4) were used to analyze the large-scale climate influences on drought and extreme precipitation in China. The study evaluated the presence of a statistically significant relationships between SPI/extreme precipitation days and EASMI/Nino 3.4 by simple linear regression. Significant relationships between drought, extreme precipitation days, and climate indices were obtained by Pearson correlation coefficient. It is noted that climate indices may influence drought and precipitation extremes of the current year and the next year [51], but after our observation, the correlation was not significant one year ahead (only 6% of the stations passed the significance test). Therefore, the study used EASMI with zero years ahead and Nino 3.4 based on the average in March the previous year to February in the following year as the candidate predictor variables. Significant relationships were supported when the correlation coefficient differed from zero in the 10% confidence interval [52].

## 3. Results

*3.1. Trends in Drought and Extreme Precipitation*

### 3.1.1. Drought Trend

Mann–Kendall test was used to analyze the trends in SPI based on monthly precipitation data from 2000 to 2015. In Figure 2b, the red dots represent relatively negative trends in terms of the SPI coefficient, i.e., clear increases in the drought trend over time. The increasing drought trend was mainly concentrated in the central and western parts of Southwest China, the northern part of East China, and Central China. The lowest SPI trend was recorded at Dali Meteorological Station in Yunnan Province. Among the 652 meteorological stations, the drought index exhibited an increasing trend at 34.5% of the stations, and there was a decreasing drought trend in most parts of the country.

### 3.1.2. Extreme Precipitation

Mann–Kendall test was used to analyze the trend of extreme precipitation based on daily precipitation data recorded in China over 16 years. From 2000 to 2015, the extreme precipitation days tended to increase in Northeast China, North China, the northeastern part of Southwest China, the western part of South China, and most of East China (the green dots in Figure 3b indicate increases in extreme precipitation). There were obvious increasing extreme precipitation trends in most of the Northeast China, and the central part of East China, the most obvious of which was recorded at Ziyang Meteorological Station in Sichuan Province. The extreme precipitation tended to increase at 51.7% of the stations, indicating an increasing trend in most parts of the country due to global warming.

### 3.1.3. Drought and Extreme Precipitation

Meteorological stations with increasing trends in terms of both regional drought and extreme precipitation were selected to characterize the superposition effect in space (red dots in Figure 4). The areas with increases in SPI and extreme precipitation were concentrated mainly in the Yangtze River Basin, especially the middle and upper reaches of the Yangtze River. This special phenomenon could possibly be explained by an energy balance climate model [53]. According to the simulation [54], wet regions get wetter and dry regions drier in the well-watered land. The Yangtze River is the boundary between dry and wet, and due to its ample water vapor conditions, it would get wetter and drier too.

$$\delta(P - E) = \alpha \delta T(P - E) \tag{3}$$

where, at temperatures typical of the lower troposphere, $\alpha \cong 0.07K^{-1}$, T is temperature, P is precipitation, and E is evaporation.

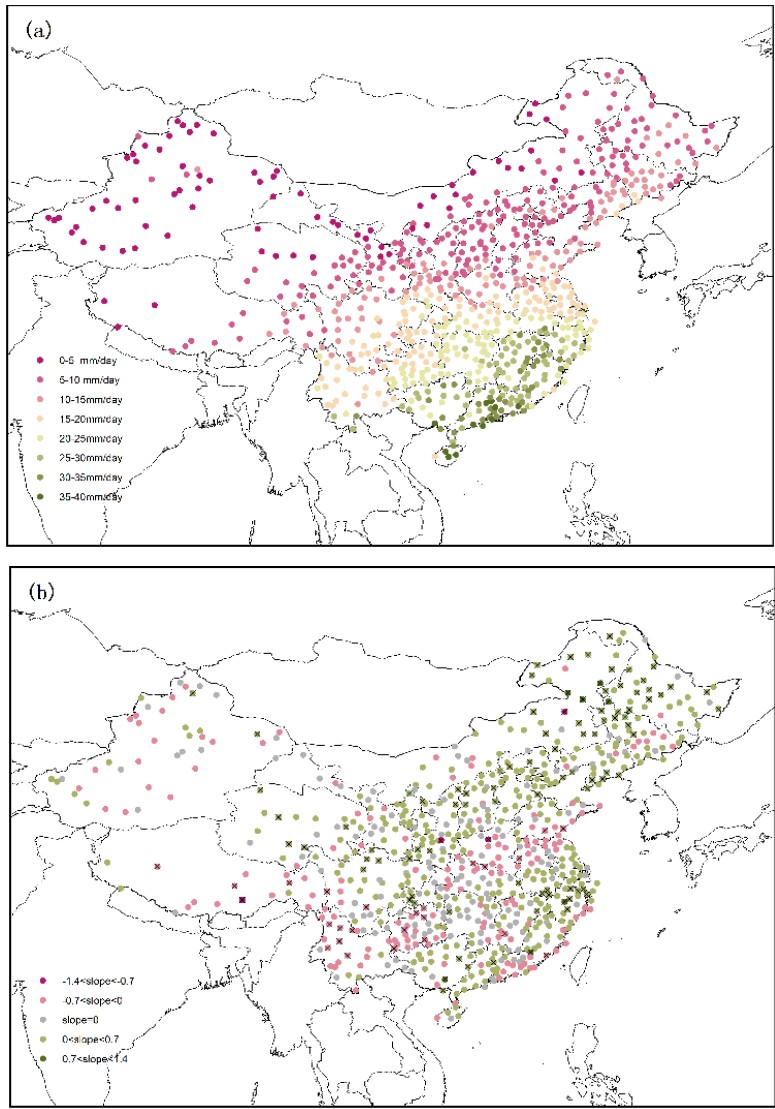

**Figure 3.** Extreme precipitation distribution: (**a**) annual average extreme precipitation days; (**b**)trend of extreme precipitation days (unit: d year$^{-1}$). Red dots indicate that the slope is negative and the extreme precipitation days tend to increase; green dots indicate that the slope is positive and the extreme precipitation days tend to decrease. The point of the cross passed the 95% significance test.

### 3.1.4. Annual Average Precipitation Trends in Subarea

Table 1 shows the annual average precipitation trends in each subarea. The annual average precipitation of each subarea increased, and East China had the largest precipitation trend, while precipitation in North China and Southwest China had only a slight increase. It is worth noting that the annual precipitation in Northwest China is increasing and has passed the 95% significance test.

**Table 1.** Annual average precipitation trends in subarea. Bold represents 95% significance test.

| Subregion | Slope |
|---|---|
| Northeast China | 0.76237 |
| North China | 0.099367 |
| East China | **2.5051** |
| South China | 1.5635 |
| Central China | 1.5194 |
| Northwest China | **1.1433** |
| Southwest China | 0.075235 |

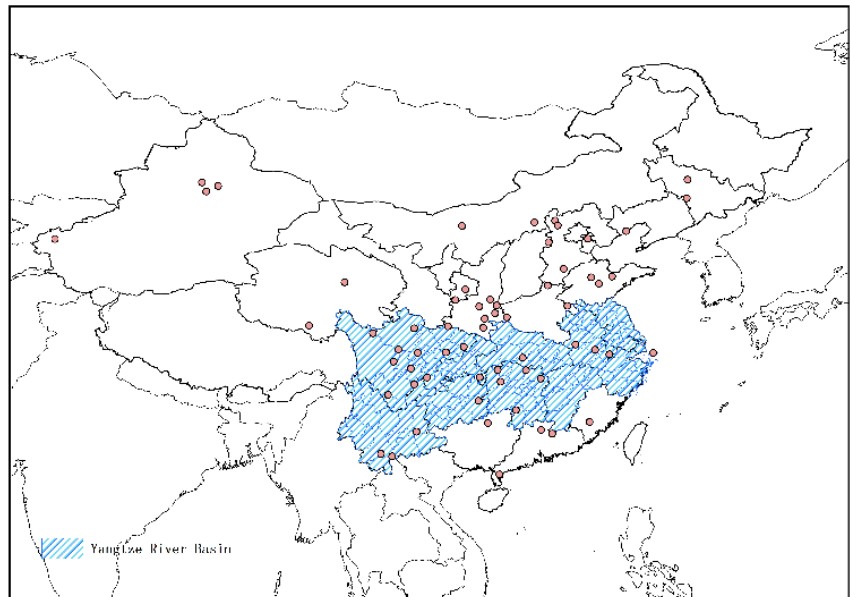

**Figure 4.** Superimposed distributions of drought and extreme precipitation. Red dots indicate stations with increased drought and extreme precipitation.

### 3.2. Correlation Analysis

The Yangtze River Basin is an important agricultural area in China and a region with rapid economic development. However, obvious changes have occurred in annual precipitation during the flood season in the Yangtze River valley, where drought and flood disasters are more frequent [55–57]. Climate disasters often cause great damage and losses to local industry and agriculture, as well as affect the safety of people and property. According to Section 3.1, the frequency of drought and floods has increased in recent years in the Yangtze River Basin. Therefore, the study focused on the correlation between the Yangtze River Basin and the climate index.

Figure 5 shows the correlations between drought, extreme precipitation frequency, and the two climate indices, EASMI and Nino 3.4. In general, only a few stations had significant correlations with EASMI and Nino 3.4. However, some spatial characteristics still could be identified. SPI was positively correlated with EASMI with zero years ahead, mainly in the south of the Yangtze River, and negatively correlated in the north of the Yangtze River, indicating that the frequency of drought in north of the Yangtze River will increase when the East Asian summer monsoon strengthens. In the northern Yangtze River, SPI and Nino 3.4 were significantly positively correlated, which indicates that when the sea surface temperature (SST) decreases in the tropical eastern Pacific, drought will increase in the northern Yangtze River, i.e., drought will occur in La Nina years. It is worth noting that the SST in the tropical eastern Pacific will affect most northern regions. In the southern part of the Yangtze River Basin, extreme precipitation days were significantly positively correlated with EASMI. Thus, extreme precipitation will increase in the southern part of the Yangtze River Basin when the East Asian summer monsoon increases. By contrast, the extreme precipitation days were significantly negatively correlated with EASMI in the northern Yangtze River, so when the East Asian summer monsoon increases, extreme precipitation will weaken in this area. The extreme precipitation days were significantly positively correlated with Nino 3.4 in most areas of the Yangtze River Basin, indicating that extreme precipitation will increase in the basin when the SST in the tropical eastern Pacific rises. Thus, extreme precipitation will increase in the Yangtze River Basin during El Niño years.

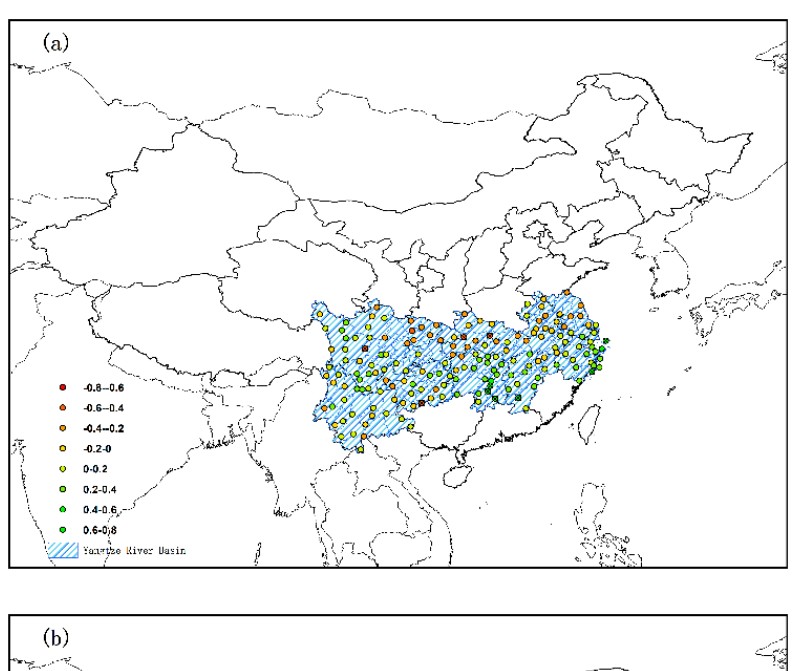

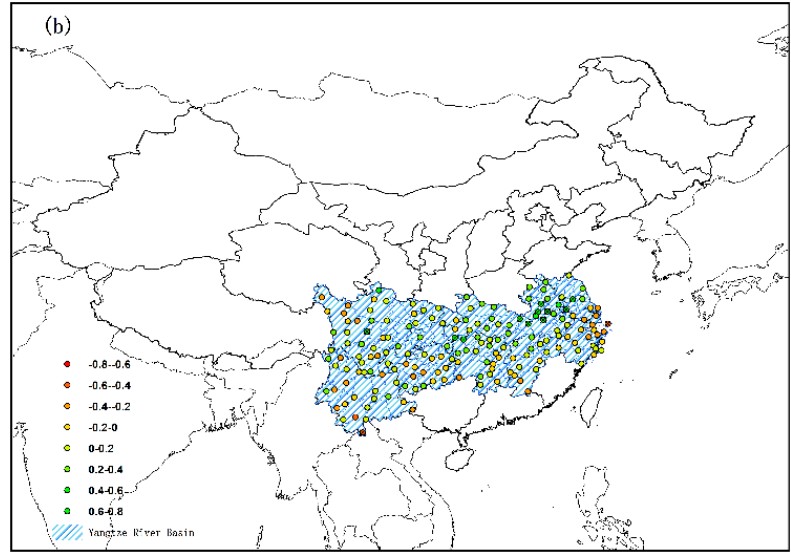

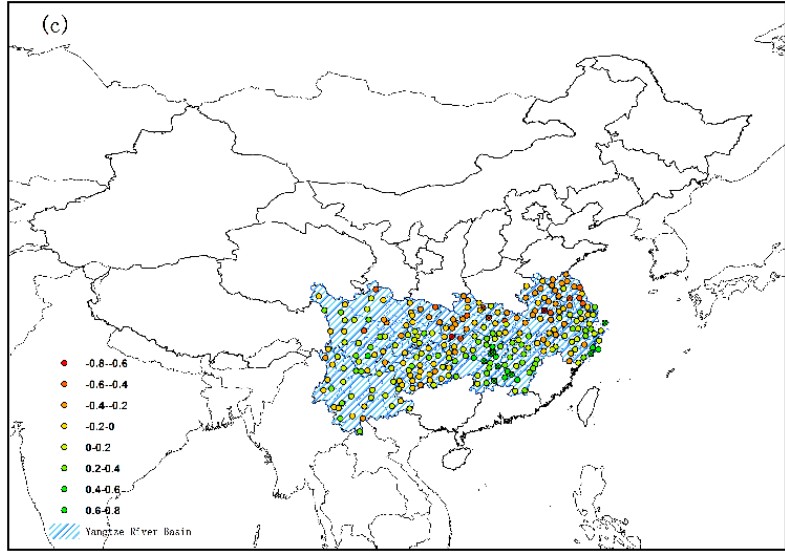

**Figure 5.** *Cont.*

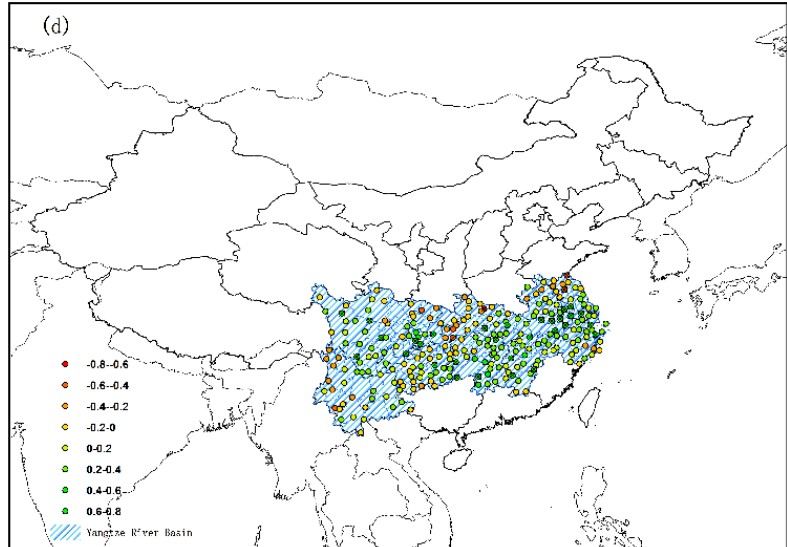

**Figure 5.** Statistical correlations between SPI, extreme precipitation days, and the East Asian Summer Monsoon Index (EASMI) and Nino 3.4. Correlations between: (**a**) SPI and EASMI, (**b**) SPI and Nino 3.4, (**c**) extreme precipitation frequency and EASMI, (**d**) extreme precipitation frequency and Nino 3.4. The point of the cross passed 90% significance test.

### 3.3. Disaster Analysis

### 3.3.1. Spatial and Temporal Distributions of Drought Losses

Figure 6 shows the EOF first mode distribution for the drought direct economic loss anomaly field in China during the 16-year period. The contribution of the first mode's variance was 36.9%, which reflects the main characteristics of the spatiotemporal changes in direct economic losses. The first dimension of the direct economic loss anomaly exhibited a downward trend from northeast to southwest, indicating that drought economic losses decreased gradually in that direction with high-value areas located in the southeast of Northeast China and north of East China (average direct economic losses amounted to RMB 13.8 billion and accounted for 18.6% of all regions). During midsummer, the day is often longer than the night in Northeast China, and the duration of sunshine is long, resulting in high evaporation and rapid onset of drought. In the spring, there is less precipitation in the Northeast China, and low precipitation but high evaporation in the summer, leading to continuous drought. After 2000, droughts occurred frequently in Northeast China and lasted for a long time. There were two consecutive dry periods in 2000–2002 and 2007–2008. In terms of the spatial distribution, 2000–2010 had the period with the highest frequency and effects of drought in Northeast China, especially in the central and western parts, where drought frequency reached 42.86% and 33.34%, respectively [58]. In addition, a comparison of the first EOF pattern in the area affected by drought in Figure 7 shows that the spatial distribution of damaged agricultural areas was roughly consistent with that of direct economic losses, which decreased gradually from the northeast to the southwest (high-value areas were located in Northeast China and Inner Mongolia, with an average damaged area of 6.44 million hectares, accounting for 31.6% of the total), thus agricultural losses accounted for a large proportion of direct economic losses. Comparing Heilongjiang and Shandong, it can be found that Heilongjiang's damaged areas were severe while the economic losses were low, and Shandong has the opposite. Shandong is a large agricultural province and its agricultural output is higher than that of Heilongjiang. Therefore, the economic losses could be high even when a drought was not particularly severe.

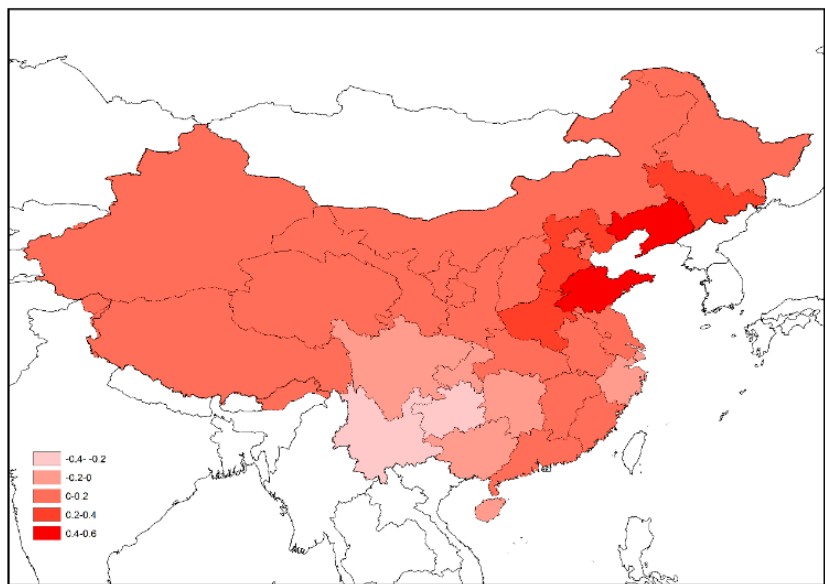

**Figure 6.** First mode of empirical orthogonal function (EOF) for the drought direct economic loss anomaly field.

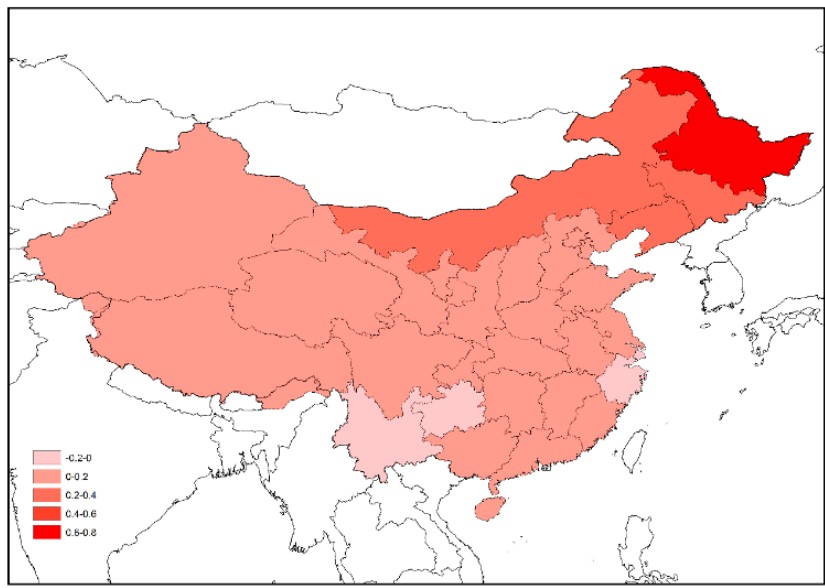

**Figure 7.** First mode of EOF for the drought damaged areas anomaly field.

The time coefficient of direct economic losses due to drought in Figure 8a show that the losses in 2000–2003, 2007, 2009, and 2014 were all greater than the average, and losses continued to increase in 2005–2007. In the drought damaged areas (Figure 8b), the losses in 2000–2004, 2006–2007, and 2009 were all greater than the average, and the area affected by disasters increased continuously in 2005–2007. The two graphs in Figure 8 show that the change in economic losses was generally consistent with the change in the affected area. However, 2003 and 2014 were rather exceptional. In 2003, the damaged areas were relatively large, but the economic losses were relatively small. By contrast, in 2014, the affected area was small, and the economic losses were relatively large. These findings may be explained by the low contribution of primary industry to gross domestic product (GDP) in 2003 and high contribution in 2014.

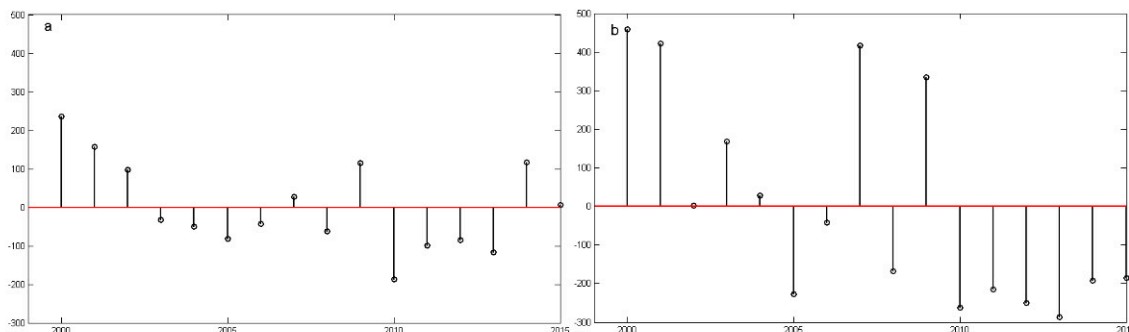

**Figure 8.** Time coefficient of (**a**) drought direct economic loss anomaly and (**b**) drought damage areas anomaly.

### 3.3.2. Spatial and Temporal Distributions of Flood Losses

Figure 9 shows the EOF first mode distribution for the flood direct economic loss anomaly field in China during the 16-year period. The contribution of the first mode's variance was 56.8%, which reflects the main characteristics of the spatiotemporal changes in the direct economic losses. Figure 9 clearly shows the abnormal flood losses in southeast of Northeast China, northeast of Southwest China and Central China (average direct economic losses amounted to RMB 56.7 billion and accounted for 51.1% of all regions). However, compared with Figure 10, the anomalous flood-damaged areas were larger in the northeast of Southwest China, north of Central China and East China (with an average damaged area of 3.97 million hectares, accounting for 43% of the total). These anomalous results may be explained by the inadequate defenses against storms and floods in Northeast China and southwestern parts of East China, whereas the central areas of East China and northern parts of Central China were more resistant to heavy rainfall. In addition, these results reflect the fact that direct economic losses caused by floods were due to other economic losses and agricultural losses. "Managing the Risks of Extreme Events and Disasters to Advance Climate Change Adaptation," issued by the Intergovernmental Panel on Climate Change specifically states the impacts of extreme and non-extreme events, and whether they constitute disasters depends on the level of exposure and vulnerability as well as the strength of the event itself. Exposure and vulnerability are key aspects when assessing the risk of disaster and its impact. The top 10 provinces in terms of annual multiyear average disaster exposure were Hubei, Anhui, Hunan, Henan, Jiangsu, Heilongjiang, Jilin, Jiangxi, Sichuan, and Shandong, as shown in Figures 9 and 10. The extremely severe disaster losses in Sichuan, Jilin, and Jiangxi may also have been caused by the high disaster exposure levels of these provinces.

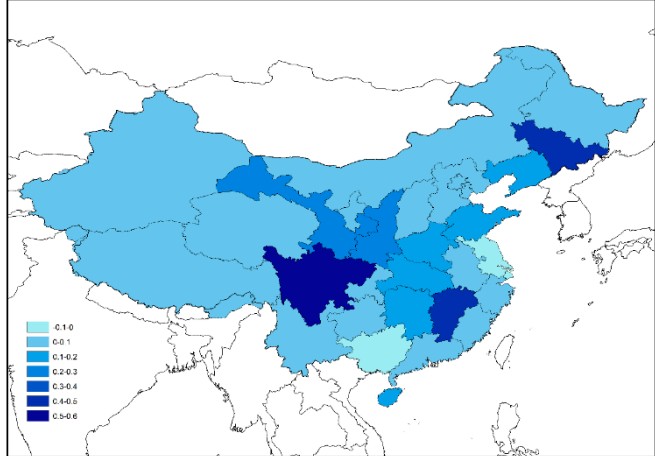

**Figure 9.** First mode of EOF for the flood direct economic loss anomaly field.

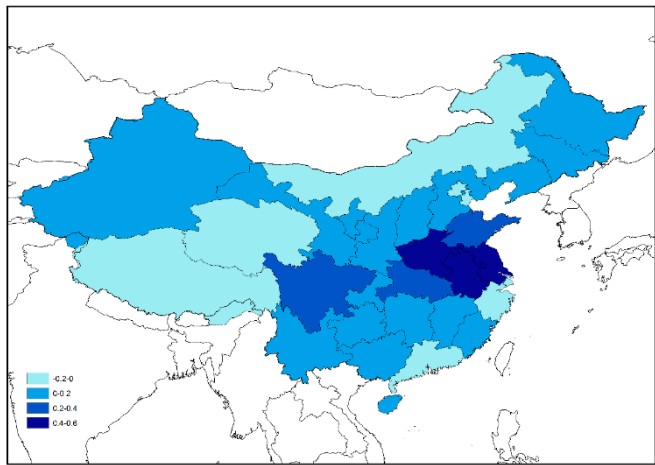

**Figure 10.** First mode of EOF for the flood-damaged areas anomaly field.

The time coefficient charts in Figure 11 show that 2010 was a year with particularly severe flooding disasters, including flood-damaged areas and direct economic losses. However, the maximum area affected by rainstorms and floods occurred in 2003 during the study period, but the direct economic losses were not severe. The reason for this difference is explained in Section 3.3.1.

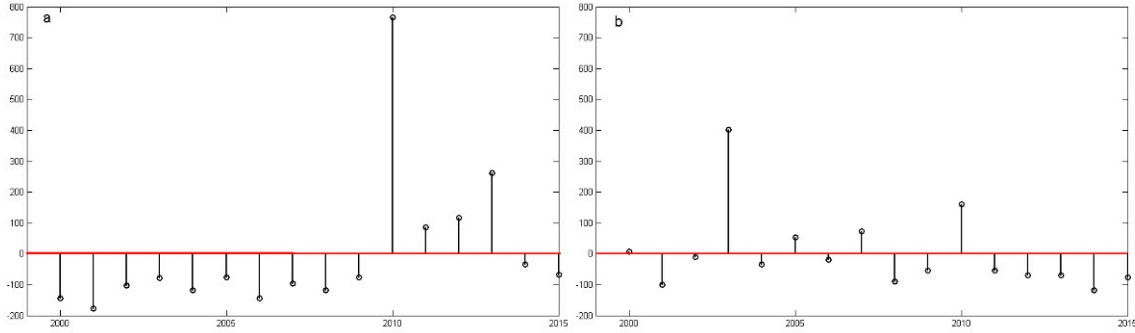

**Figure 11.** Time coefficient of (**a**) flood direct economic loss anomaly and (**b**) flood-damaged area anomaly.

## 4. Summary and Discussion

Based on the results obtained in this study can give the following main conclusions.

(1) 34.5% and 51.7% of all the stations over China have an increasing tendency in drought and flood, respectively.

(2) Special climate phenomena have been observed in the Yangtze River Basin due to its special circulation characteristics and geographical location. Meteorological disasters have occurred frequently in recent years in the basin, where the frequency of both drought and extreme precipitation has increased (Figure 4).

(3) A comparison of the drought trend distribution (Figure 2b) and drought damaged areas (Figure 7) shows that the drought damaged areas were relatively severe in the north of East China and Central China, and drought will increase in these areas. Therefore, it is necessary to increase defense and control measures to prevent drought disasters in these regions.

(4) A comparison of the extreme precipitation trend distribution (Figure 3b) and flood-damaged areas (Figure 10) shows that flood-damaged areas were relatively severe in the southern part of East China, and extreme precipitation also tend to increase in most of this area. Therefore, it is necessary to increase defense and control measures to prevent flood disasters in this region.

(5) Northeast China has been greatly affected by drought, but the drought trend is weakening (Figure 2), whereas extreme precipitation (Figure 3) has tended to increase. Thus, extreme precipitation has shifted to the north with less drought, and the frequency of extreme precipitation may increase. However, the north of East China and Central China have been affected by severe floods, but the drought trend is increasing (Figure 2) and extreme precipitation (Figure 3) is decreasing, indicating that drought is shifting to the south with fewer floods, but there is the possibility of more severe drought.

Climate variability can play a significant role in explaining the variations in frequency and intensity of extreme events. The findings show the challenges of water resource management in a changing climate. The study combined SPI and extreme precipitation days to study extreme meteorological events and explore the interaction between climate change factors and socio-economic factors in this study. However, considering these two indices alone does not provide an in-depth understanding of the changes in extreme events. To better respond to climate change and ensure sustainable economic development, more research tools need to be added in future work, such as the Standardized Precipitation–Evapotranspiration Index (SPEI). The SPEI considers not only precipitation but also evapotranspiration data in its calculation, allowing for a more complete approach to exploring the effects of climate change on drought conditions. Therefore, there are still challenges waiting for us to solve.

**Author Contributions:** conceptualization, J.C. and W.D.; methodology, T.X.; software, T.X.; validation, T.X.; formal analysis, T.X.; investigation, J.C. and T.X.; resources, J.C. and T.X.; data curation, Y.X.; writing—original draft preparation, T.X.; writing—review and editing, J.C. and T.X; visualization, T.X.; supervision, T.X.; project administration, J.C.; funding acquisition, J.C.

**Funding:** This work was supported by National Key Research and Development Program of China (grant 2018YFC1509003), National Key Research and Development Program of China (grant 2016YFA0602703), National Natural Science Foundation of China (grant 41575001), Skate Key Laboratory of Earth Surface Processes and Resource Ecology Project (2017-FX-03), and Supported Scientific Research Foundation Beijing Normal University (2015KJJCA14).

**Acknowledgments:** Thanks to the "International Science Editing" for helping us to polish this article.

**Conflicts of Interest:** No conflict of interest exits in the submission of this manuscript, and manuscript is approved by all authors for publication.

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
