# Peer review of "Regional Temporal and Spatial Trends in Drought and Flood Disasters in China and Assessment of Economic Losses in Recent Years"

_sustainability, doi:10.3390/su11010055_

Reviewer 1 Report

I'd like to suggest three following points which you need to reply.

In 2.4.2. Mann-Kendall test section, you need to revise the first sentence, because that lacks verb,

In 3.1.2 Extreme precipitation section, it's better to indicate a concrete station name with most severe impact as same as 3.1.1,

In conclusion part, you need to identify future challenges including using the Standardized Precipitation-Evapotranspiration Index (SPEI) instead of SPI.

Author Response

Dear reviewer:

We are very grateful to your comments for the manuscript. According with your advice, we amended the relevant part in manuscript. Some of your questions were answered below.

Point 1: In 2.4.2. Mann-Kendall test section, you need to revise the first sentence, because that lacks verb,

Response 1: Thanks a lot for pointing out the grammar mistake. After careful consideration, we have revised the first sentence in Section 2.4.2 as follows:

The nonparametric Mann–Kendall (MK) test was used to characterize the trends of SPI and extreme precipitation days in this study.

Point 2: In 3.1.2 Extreme precipitation section, it's better to indicate a concrete station name with most severe impact as same as 3.1.1,

Response 2: Thank you very much for your suggestion. We have added the station with most severe impact in 3.1.2, Ziyang Meteorological Station in Sichuan Province.

Point 3: In conclusion part, you need to identify future challenges including using the Standardized Precipitation-Evapotranspiration Index (SPEI) instead of SPI.

Response 3: Your suggestion is very helpful to us. We have included future challenges in the revised Conclusions and Discussions section as you suggested.

Reviewer 2 Report

The aim of the paper is to better understand the distribution of floods and droughts in mainland China, and quantify the economic losses. The authors use a range of methods including statistical analyses, observational and trends data to quantify impacts.

I note in the acknowledgements that the manuscript appears to have been edited by ‘International Science Editing’, however it is still very difficult to follow, and should be substantially revised.
The abstract should be rewritten. Lines 12 and 13, the sentence is unclear. How does understanding distribution (spatial distribution? Temporal distribution?) play an important role in disaster risk management?

Line 15 – “We found…”. Avoid personal pronouns in scientific writing. “The study found…” “The research found…”

Line 22 “… the Southwest China, where accompanied by severe disaster losses…” – do the authors mean ‘were accompanied by severe…’ or if it is ‘where’, then the sentence should rewritten to make that more clear.

Line 34 – region and the [sic], should just be ‘region and Southwest China’

Line 36 – water vapor, no need for capitalisation

Line 37 – Drought will be more likely TO occur…

Line 43 – 1950 – 2008 (?) Why does this reference not include more recent (e.g AR5, 2014 or any of the published (2018) literature? – has ‘recent global warming’ ceased? Again, this should be updated.

Line 57 – According to recent statistics… - which recent statistics? Reference please.

I do apologise, however the paper – while interesting, does need to be substantially revised in order to make it readable.

I would suggest the authors use a reputable editor, or academic support services to rewrite the manuscript, paying particular attention to their discussion of the results, and please re-write a proper conclusion (ie. not a bulleted list).

The topic is interesting, and worthy of study – however I am unable to comment further on the manuscript until it has been substantially revised.

I am sorry that I am unable to provide a stronger recommendation for this paper. There is an interesting piece of research here, and as a scientist working quite closely at the interface of science and policy, these sorts of case studies are valuable – and necessary.  However, the paper must be substantially rewritten to address these issues.

I hope the authors take the opportunity to rewrite the manuscript and resubmit at a future date.

Author Response

Dear reviewer:

Many thanks for your advice and time

We feel encouraged that you found the topic interesting and worthy of study. Other reviewers also approved the value of our work.

As for the language problem, we have followed your advice to use the editing service recommended by Sustainability in order to improve the manuscript’s readability. In addition, the Conclusion part is rewritten as the Discussion and Conclusion part with more details and deeper reflection on the research and its implications.

We want to thank you again for your patient and helpful suggestions. It would be very kind of you to review our substantially revised version soon.

The detailed response to your comments is as follows, in case you are still concerned.

Point 1: The abstract should be rewritten. Lines 12 and 13, the sentence is unclear. How does understanding distribution (spatial distribution? Temporal distribution?) play an important role in disaster risk management?

Response 1: We have rewritten the abstract to make it clear. Distribution refers to temporal and spatial distribution as addressed in line 12.

Point 2: Line 15 – “We found…”. Avoid personal pronouns in scientific writing. “The study found…” “The research found…”

Response 2: Thanks for pointing out the personal pronoun problem. We have revised it in line 17 and adjusted similar expressions within the whole manuscript. We will pay more attention to the academic writing norm in the future.

Point 3: Line 22 “… the Southwest China, where accompanied by severe disaster losses…” – do the authors mean ‘were accompanied by severe…’ or if it is ‘where’, then the sentence should rewritten to make that more clear.

Response 3: Thank you very much for your suggestion. We have revised it in line 26 as follows:

In the Yangtze River Basin, there are increasing trends in terms of drought and extreme precipitation, especially upstream of the Yangtze River, accompanied by severe disaster losses.

Point 4:  Line 34 – region and the [sic], should just be ‘region and Southwest China’

Response 4: We are sorry for this kind of mistakes. The sentence has been rewritten in line 43-46.

Point 5:  Line 36 – water vapor, no need for capitalisation

Response 6: Thank you very much for your suggestion. We have changed in line 50.

Point 6: Line 37 – Drought will be more likely TO occur…

Response 6: The sentence in Line 51 is edited as you suggested.

Point 7: Line 43 – 1950 – 2008 (?) Why does this reference not include more recent (e.g AR5, 2014 or any of the published (2018) literature? – has ‘recent global warming’ ceased? Again, this should be updated.

Response 7: This passage has been rewritten in line 55-68. We have added some updated references to address the global warming issue.

Point 8: Line 57 – According to recent statistics… - which recent statistics? Reference please.

Response 8: References are added in line 81.

Reviewer 3 Report

The authors present an interesting investigation related to a very relevant issue, such as the coupled effect of droughts and floods. Although both the topic and the content of the manuscript are attractive, there are some issues that need to be addressed prior to publication:

1) The levle of English of the manuscript must be improved. The authors are suggested to seek for professional help for this, since the document contains a lot of grammar and spelling errors that need to be amended.

2) Line 33. I doubt if talking about recent years is right when you are omitting almost a lustrum of analysis. Some update should be conducted in this respect.

3) The introduction lacks some figures about previous natural disasters in the form of droughts and floods across the world, highlighting the losses derived from them too in order to provide evidence of the need for tools and methods to better manage these phenomena.

4) Although the contribution of the paper is well defined, i.e. the joint evaluation of droughts and floods, the literature review provided is too weak. The authors should revise more publications in this section, in order to further strengthen the contributions of this research to the State of the Art in which is framed.

5) Figures 1 and 2 are perfectly compatible and, consequently, must be combined in a single figure.

6) Lines 151-152. This sentence is incomplete. Please, revise.

7) Line 160. There is no Pearson regression model, it is actually called Pearson correlation coefficient. Besides, I cannot find any justification to use a significance level of 0.10, apart from obtain more significant correlations. The fact that the value of 0.05 has become a synonym of significance level in any scientific discipline should mean something for the authors.

8) Lines 170-171. That same sentence was already mentioned in the methodology. Please, avoid any repetition throughout the manuscript.

9) Figures 3, 4 and similar. The legend used to represent the different variables must be changed, for the sake of visualization.

10) Line 202. The full meaning of P-E is not provided anywhere.

11) The conclusions are not properly approached. They are limited to repeat the most relevant results stemming from section 3. Instead, they should address the implications of this research for the community, the future lines of investigation derived from this and the limitations of the proposed approach. For instance, the authors could have conducted an analysis to explore how these results might vary under different Climate Change scenarios (RCPs)

Author Response

Dear reviewer:

We are very grateful to your comments for the manuscript. According with your advice, we amended the relevant part in manuscript. Some of your questions were answered below.

Point 1: The levle of English of the manuscript must be improved. The authors are suggested to seek for professional help for this, since the document contains a lot of grammar and spelling errors that need to be amended.

Response 1: We are sorry for the errors in the manuscript. We have submitted the manuscript to MDPI for English editing to increase the readability of the manuscript.

Point 2:  Line 33. I doubt if talking about recent years is right when you are omitting almost a lustrum of analysis. Some update should be conducted in this respect.

Response 2: Thank you very much for your suggestion. We have revised and update this analysis in line 43 after reading more references.

Point 3: The introduction lacks some figures about previous natural disasters in the form of droughts and floods across the world, highlighting the losses derived from them too in order to provide evidence of the need for tools and methods to better manage these phenomena.

Response 3: Your suggestion is very helpful to us. We have added some research data from WMO in Introduction like line 37 to provide evidence of the need for tools and methods to better manage disasters.

Point 4: Although the contribution of the paper is well defined, i.e. the joint evaluation of droughts and floods, the literature review provided is too weak. The authors should revise more publications in this section, in order to further strengthen the contributions of this research to the State of the Art in which is framed.

Response 4: Thank you very much for your suggestion. More publications are provided in the text and references to strengthen the contribution of this research.

Point 5:  Figures 1 and 2 are perfectly compatible and, consequently, must be combined in a single figure.

Response 5: We have followed your good advice to combine the previous fig.1 and fig.2 to the new fig. 1 in the manuscript.

Point 6:   Lines 151-152. This sentence is incomplete. Please, revise.

Response 6: Sorry for our mistake. We have revised it in line 179-180 as follows:

The nonparametric Mann–Kendall (MK) test was used to characterize the trends of SPI and extreme precipitation days in this study.

Point 7:  Line 160. There is no Pearson regression model, it is actually called Pearson correlation coefficient. Besides, I cannot find any justification to use a significance level of 0.10, apart from obtain more significant correlations. The fact that the value of 0.05 has become a synonym of significance level in any scientific discipline should mean something for the authors.

Response 7: We are sorry that due to the mother tongue, we have made a wrong statement about the Pearson correlation coefficient, and we have revise it in Line 189.

As for the issue of significance test, we have made changes in the figures of lines 284 to 291. In order to better represent the relationship between drought and flood with climate index in the Yangtze River Basin, we put all the stations of this area in the figures. The stations that pass the 90% significance test are marked in the figures. But we think that a significant level of 0.1 may be feasible, because we have seen some studies in some references like Deng (2018) and Mallakpour (2016). If we can, we want to continue to use the level of significance of 0.1.

Point 8:  Lines 170-171. That same sentence was already mentioned in the methodology. Please, avoid any repetition throughout the manuscript.

Response 8: We are sorry for the repetition. We will try to avoid repeating again and we have revised it in the Lines 200-201.

Point 9: Figures 3, 4 and similar. The legend used to represent the different variables must be changed, for the sake of visualization.

Response 9: Thanks for your good advice. We have changed the legend in figures 3,4.

Point 10: Line 202. The full meaning of P-E is not provided anywhere.

Response 10: The pattern of P - E is an energy balance climate model. We have added a supplementary explanation in line 242 to enhance the readability of this manuscript.

Point 11: The conclusions are not properly approached. They are limited to repeat the most relevant results stemming from section 3. Instead, they should address the implications of this research for the community, the future lines of investigation derived from this and the limitations of the proposed approach. For instance, the authors could have conducted an analysis to explore how these results might vary under different Climate Change scenarios (RCPs)

Response 11: Your suggestion is very helpful to us. We have changed the Conclusions section to Conclusions and Discussions section and added future challenges including using the SPEI in it.

Our next work will add precipitation research in different climate change scenarios, but in order to better fit the theme of this manuscript, we intend to do related research later.

References

Deng, Y., Jiang, W. G., He, B., Chen, Z., Jia, K. Change in Intensity and Frequency of Extreme Precipitation and its Possible Teleconnection With Large‐Scale Climate Index Over the China From 1960 to 2015. J. Geophys. Res-Atmos 2018, 123 (4),

Reviewer 4 Report

General comments

The study represents an interesting attempt to evaluate the impact of increasing extreme events related to climate change in China. The authors evaluated the impact of increased frequency and intensity of both floods and droughts, by relating trends of extreme events with relevant climate indices (East Asian Summer Monsoon Index -EASMI and the Nino 3.4 index) and induced economic losses.

The manuscript is well structured and appropriately written; the study is correctly designed and the results appropriately interpreted for supporting their conclusions. Some minor revisions are required to make it more readable.

The article is interesting for a broad sustainability audience involved in studies on the impacts of climate change in addition to the direct interest for Chinese policy makers for identifying and managing measures to mitigate the risk induced by the increase in frequency of extreme events.

Specific comments

Lines 76-79: “Daily and monthly precipitation records from 855 stations in China during 2000–2015 were used” …. “Relatively strict quality control was applied before use and stations with short or missing data sequences were deleted”. Throughout the manuscript are always reported 855 stations for the analyses (see, e.g., line 91, line 175). How many stations were discarded/deleted ?

Line 94-95: Generally, SPI between -1 and +1 is considered as an ordinary variability.

Line 151: “In this paper, the Mann-Kendall test, which is widely used to detect trends in meteorological data [35].” … Hanging sentence, please modify.

Line 161-164. The authors indicate that the correlation between climate indices and drought/extreme precipitation days was not significant with one year of shift. Please add information on such a test (e.g., add correlation coefficient per regions, or number of series with p>0.1).

Section 3.1.2: Please add a table with information on n. of stations with positive and negative trends per sub-region, n. of stations with positive and negative trends passed 95% significance test.

Section 3.1.2: I would be also interesting to show the temporal profile 2000-2015 per sub-region to better understand the periods driving the trends. The author can add the mean values for positive and negative trends for each sub-region.

Lines 201-204: Please better explain the concept on Yangtze River effect (… wetter and drier too ???)

Line 217-218: Please add value for significance (p> …).

Figures: Please increase the character of numerical legends (e.g. Figure 4, Figure 6); they are not clearly readable.

Author Response

Dear reviewer:

We are very grateful to your comments for the manuscript. According with your advice, we amended the relevant part in manuscript. Some of your questions were answered below.

Point 1: Lines 76-79: “Daily and monthly precipitation records from 855 stations in China during 2000–2015 were used” …. “Relatively strict quality control was applied before use and stations with short or missing data sequences were deleted”. Throughout the manuscript are always reported 855 stations for the analyses (see, e.g., line 91, line 175). How many stations were discarded/deleted ?

Response 1: We are very sorry for our mistake on the stations. 203 stations were deleted in our manuscript and we have revised in line 117 and 205.

Point 2:  Line 94-95: Generally, SPI between -1 and +1 is considered as an ordinary variability.

Response 2: We have revised the value of SPI in line 120,121 as you suggested

Point 3: “In this paper, the Mann-Kendall test, which is widely used to detect trends in meteorological data [35].” … Hanging sentence, please modify.

Response 3:  Sorry for the mistake. We have revise it in line 179-180.

Point 4: Line 161-164. The authors indicate that the correlation between climate indices and drought/extreme precipitation days was not significant with one year of shift. Please add information on such a test (e.g., add correlation coefficient per regions, or number of series with p>0.1).

Response 4: Thanks for your suggestion. According to our observation, only 6 of the stations passed the significance test. We have added the information in line 191-192.

Point 5: Section 3.1.2: Please add a table with information on n. of stations with positive and negative trends per sub-region, n. of stations with positive and negative trends passed 95% significance test.

Response 5: After careful considering your comments, we added a table in Section 3.1.4, which shows the annual average precipitation trends in each sub-area. We hope this form reflects the difference in precipitation in different regions.

Point 6:  Section 3.1.2: I would be also interesting to show the temporal profile 2000-2015 per sub-region to better understand the periods driving the trends. The author can add the mean values for positive and negative trends for each sub-region.

Response 6: We have added a table on line 254 to reflect the variation characteristics of precipitation in different regions during the study period.

Point 7:  Lines 201-204: Please better explain the concept on Yangtze River effect (… wetter and drier too ???)

Response 7: We have added a supplementary explanation in line 238-245 to enhance the readability of this phenomenon.

Point 8:  Line 217-218: Please add value for significance (p> …).

Response 8: We have shown the correlation of all the stations in the Yangtze River Basin and identified the stations through the significance test in line 284-290.

Point 9: Figures: Please increase the character of numerical legends (e.g. Figure 4, Figure 6); they are not clearly readable.

Response 9: We have changed the legend in figures 2,3,5 to make them readable as you suggested.

Point 10: Line 202. The full meaning of P-E is not provided anywhere.

Response 10: The pattern of P - E is an energy balance climate model. We have added a supplementary explanation in line 242 to enhance the readability of this manuscript.

Round  2

Reviewer 2 Report

The manuscript is much improved, and I am pleased to see the changes.

Author Response

Comment: The manuscript is much improved, and I am pleased to see the changes.

Response:

Thanks reviewer for good comments and hard work. Your comments encourage us to continue our research. 

Reviewer 3 Report

The authors are comended for their efforts to improve the manuscript. However, it keeps lacking a proper literature review related to the modelling of floods and droughts, either standalone or in combination. Also, although the conclusions have been improved in comparison with the original version of the manuscript, references should not been added in this section.

Author Response

Dear reviewer:

We are very grateful to your comments for the manuscript. According with your advice, we amended the relevant part in manuscript. Your questions were answered below.

Point 1: The authors are comended for their efforts to improve the manuscript. However, it keeps lacking a proper literature review related to the modelling of floods and droughts, either standalone or in combination. Also, although the conclusions have been improved in comparison with the original version of the manuscript, references should not been added in this section.

Response 1: Many thanks for your advice and time

We added some literatures related to the modelling of floods and droughts in line 69-7588,89 after considering the structure of the introduction comprehensively.

After our careful consideration, your suggestion is reasonable for our manuscript. We have made some changes to the discussion and conclusions and removed unnecessary references.
